Parametric art creation platform design based on visual delivery and multimedia data fusion

Yun Qing yun2357245@163.com
School of International Exchange, Kyungil University , Gyeongsan-s, Gyeongsangbuk-do , Republic of South Korea
Sohaib Osama
Electronic publication date: 2025 Sep 16
Publication date: 2025
Volume: 11
Electronic Location ID: e3175
Received 2025 May 5; Accepted 2025 Aug 7
Copyright: © 2025 Yun
Copyright year: 2025
Copyright holder: Yun
License: This is an open access article distributed under the terms of the Creative Commons Attribution License, which permits unrestricted use, distribution, reproduction and adaptation in any medium and for any purpose provided that it is properly attributed. For attribution, the original author(s), title, publication source (PeerJ Computer Science) and either DOI or URL of the article must be cited.
License URL: https://creativecommons.org/licenses/by/4.0/

Keywords: Deep Q-network (DQN), Feature fusion, Visual communication, Long short-term memory (LSTM), Fully convolutional network (FCN)

Funding: The author received no funding for this work.

==============================
In the era of informational ascendancy, the discourse of artistic communication has transcended the confines of conventional physical domains and geographical boundaries, extending its purview ubiquitously across the global expanse. Consequently, the predominant mode of artistic interaction has evolved towards swift and extensive engagement through virtual platforms. However, this paradigm shift has given rise to the imperative task of meticulous categorization and labeling of an extensive corpus of artistic works, demanding substantial temporal and human resources. This article introduces an innovative bimodal time series classification model (BTSCM) network for the purpose of categorizing and labeling artworks on virtual platforms. Rooted in the foundational principles of visual communication and leveraging multimedia fusion technology, the proposed model proves instrumental in discerning categories within the realm of video content. The BTSCM framework initiates the classification of video data into constituent image and sound elements, employing the conceptual framework of visual communication. Subsequently, feature extraction for both forms of information is achieved through the application of Inflated 3D ConvNet and Mel frequency cepstrum coefficient (MFCC). The synthesis of these extracted features is orchestrated through a fusion of fully convolutional network (FCN), deep Q-network (DQN), and long short-term memory (LSTM), collectively manifesting as the BTSCM network model. This amalgamated network, shaped by the union of fully convolutional network (FCN), DQN, and LSTM, adeptly conducts information processing, culminating in the realization of high-precision video classification. Experimental findings substantiate the efficacy of the BTSCM framework, as evidenced by outstanding classification results across diverse video classification datasets. The classification recognition rate on the self-established art platform exceeds 90%, surpassing benchmarks set by multiple multimodal fusion recognition networks. These commendable outcomes underscore the BTSCM framework’s potential significance, providing a theoretical and methodological foundation for the prospective scrutiny and annotation of content within art creation platforms.

Introduction

Visual communication, an amalgamation of art and technology, serves as a conduit for conveying information while orchestrating a nuanced visual experience through the adept utilization of visual elements encompassing images, graphics, colors, and typography. In tandem, it adheres to the tenets of design principles. This intricate discipline spans a diverse spectrum of media and domains, traversing realms such as print, digital media, advertising, brand design, film, and animation. Crucially, it stands as an indispensable theoretical framework crucial to the evolution of contemporary artistic products. The cultural paradigm shift, instigated by the information age propelled by digital technology, unfolds both opportunities and challenges for the trajectory of visual communication design in this nascent era (Anggrianto, 2022). Under the pervasive influence of digital technology, communication media and forms pivot towards novel contexts characterized by virtual dimensions, multi-dimensionality, dynamism, intelligence, and interactive design. Simultaneously, technology undergoes transformative shifts, reshaping the arsenal of design tools and methodologies. This transformative tide profoundly impacts society’s comprehension of design in its entirety. The pervasive influence of digital technology on design innovation within the information age bequeaths an expansive platform and a promising prospect for the future trajectory of visual communication design (Aiello & Parry, 2019).

Visual communication shares an intimate symbiosis with multimedia technology, and their synergistic fusion yields a more opulent and profound visual experience. Multimedia technology orchestrates the amalgamation of diverse media elements, encompassing text, images, audio, and video. Meanwhile, visual communication assumes a pivotal role, serving as a linchpin that imparts information and emotion through the nuanced interplay of visual elements and design principles. In the realm of design, the incorporation of advanced deep learning models, exemplified by generative adversarial networks (GANs), facilitates image generation, engendering realistic visuals that elevate the horizons of creativity and imagination (Dharanya, Raj & Gopi, 2021). This innovation holds promise for diverse applications, spanning advertising to the creation of artistic masterpieces. Noteworthy are models such as convolutional neural networks (CNN) within the deep learning paradigm, attaining remarkable success in the domains of image recognition and classification. In the sphere of multimedia technology, these models find utility in image search, automatic labeling, and content classification, thereby augmenting the efficiency of managing and retrieving multimedia information. Moreover, recurrent neural networks (RNN) and long short term memory (LSTM) models, specialized in sequence data processing, find apt application in diverse domains such as audio synthesis and text generation (Li et al., 2019). Hence, unleashing the full spectrum of technical prowess inherent in deep learning at this juncture emerges as the linchpin for refining the seamless integration of visual communication and multimedia.

Within the prevailing art creation platform, the conventional manual labeling technology grapples with the formidable challenge of annotating voluminous creative works. Hence, imperative is the adoption of advanced deep learning technologies to accomplish the labeling of uploaded artworks on the art platform. Leveraging potent networks like CNN, ResNet, and others in image processing proves particularly efficacious, showcasing superior performance in the domain of static image recognition and analysis (Nguyen et al., 2021). For image works uploaded on the art platform, the deployment of robust networks such as CNN and ResNet augments the platform’s analytical capabilities, particularly excelling in static image recognition and analysis (Nguyen et al., 2021). Conversely, in the realm of video motion data, characterized by multidimensional temporal intricacies and enriched with textual and auditory nuances, an enhanced analytical proficiency is achieved by delving into the time-frequency domain of audio. Techniques like spectral transform contribute significantly to ameliorating analysis capabilities while concurrently mitigating data processing dimensions. Commonly utilized audio features, including Mel frequency cepstrum coefficients (MFCC) and time-domain zero-crossing rate, emerge as pivotal tools in enhancing the performance of sound signals. In the context of visual communication, this article addresses the audit of artworks on art platforms and establishes a comprehensive work categorization network that seamlessly integrates multimedia data. The distinct contributions of this article unfold as follows:

(1) To meet the requisites of video data recognition, a segmentation of image and audio data is undertaken, culminating in the proposal of a feature extraction fusion network. This network, grounded in the I3D architecture for image feature extraction and MFCC for audio feature extraction, serves to augment the overall data performance.

(2) The extracted bimodal data features undergo a transformative integration within a BTSCM network. This model, constructed upon multiple deep learning modules, orchestrates data classification through the synergistic interplay of reinforcement learning, LSTM, and FCN. The integration of these diverse modules achieves an optimal refinement of data classification.

(3) Employing the devised BTSCM network model, video classification across diverse datasets and rigorous testing on a self-established database are executed. The empirical results substantiate the efficacy of the methodological framework proposed in this article, attesting to its exceptional performance and validating its effectiveness in real-world applications.

The rest of the article is arranged as follows: related works for multimedia data fusion are presented in ‘Related Works’. ‘Material and Methods’ introduced the feature extraction process and the BTSCM. Experiment results and related analysis is detailed described in ‘Experiment’ and ‘Discussion’ is the discussion. The conclusion is drawn at last.

Related works

Traditional data fusion methods

Traditional methodologies for data fusion primarily encompass feature-level homology and decision-level fusion. Decision-level fusion involves amalgamating results obtained from the recognition of disparate data modalities, whereas feature-level fusion entails the amalgamation of features within the feature space post feature extraction and spatial transformation of distinct modality data. The culmination of feature fusion results in the derivation of ultimate recognition outcomes. In contrast to decision-level fusion, feature fusion adeptly exploits the synergies inherent in interconnecting different modal features within the feature space. Ngiam et al. (2011) introduced a dual-pathway self-encoder model, extending the self-encoder paradigm for the fusion of two modalities. This model intricately intertwines two self-encoders, each endowed with independent encoders and decoders, connected through a shared hidden representation layer. Leveraging an encoder grounded in the restricted Boltzmann machine (Lee, Ekanadham & Ng, 2007), this design facilitates efficient interaction of features from distinct modalities at the hidden representation layer. Wang et al. (2015) further refined this model by incorporating orthogonal regularization of the weight matrix, mitigating redundancy in information. Silberer & Lapata (2014) embraced the concept of dual-pathway self-encoders to extract high-dimensional features effectively from textual and image inputs. However, none of the aforementioned dual-path autocoder models have been deployed in the domain of multimodal emotion recognition tasks. The utilization of restricted Boltzmann machines in these models imposes limitations on feature extraction capabilities, rendering the training process cumbersome. In addition to autocoders, conventional correlation analysis (Hotelling, 1992) found application in modal fusion. Originally designed for measuring correlation between two vectors, its scope was limited. Andrew et al. (2013) innovatively proposed a deep typical correlation analysis algorithm, intertwining deep neural networks with typical correlation analysis. This integration enables the application of correlation analysis to high-dimensional features, leveraging deep neural networks to learn intricate nonlinear transformations from varied inputs. Subsequent work (Dumpala et al., 2018) harnessed the nonlinear transformation capabilities of deep typical correlation analysis, coupling it with feature correlation measures for applications in sentiment recognition. Beyond dual-pathway autocoders and deep typical correlation analysis, modal fusion has also explored the utilization of mutual information.

Research on deep multimodal feature layer fusion methods

In the relentless progression of deep learning technology, the prevailing approach in contemporary data analysis involves fusing data at distinct stages by extracting features through deep networks. This paradigm is exemplified by the utilization of advanced mechanisms such as attention mechanisms and generative adversarial networks to discern subtle correlations between modalities. For instance, Zadeh et al. (2018) introduced a pioneering memory fusion network, delineating modality-specific and cross-modality interactions across the temporal axis. This model employs a distinctive attention mechanism to generalize temporal information, incorporating multimodal gated memories. In a subsequent model (Zadeh et al., 2018), the same authors explored modal interaction within a time series using a “multi-attention module” within a neural network. The interactions are stored in a hybrid memory within a recurrent memory component. Additionally, Tsai et al. (2019) proposed a fusion model grounded in a multi-attention mechanism. This model leverages different modal data as query value, key value, and value in the attention module, uncovering cross-modal interaction information within the graph computation feature of the attention module. Sahu & Vechtomova (2021) innovated with a fusion algorithm grounded in generative adversarial networks. This approach reveals commonalities among diverse modal features through an adversarial training strategy.

In delving into the study of video data, an optimal approach involves fully harnessing the distinctive characteristics of the two modal information types: image and audio. Employing advanced feature layer fusion methods facilitates the precise analysis of data, particularly in the realm of multimodal video data encompassing both image and sound. In the exploration of multimodal data, with a primary emphasis on image data, Tran et al. (2015) introduced a novel approach involving the calculation of three-dimensional convolutional neural networks, considering spatial and temporal dimensions. This model, operating through successive frames and convolution operations, adeptly amalgamates temporal information within the video domain. Convolutional neural networks, known for their prowess in processing multi-channel image information, play a pivotal role in image recognition, segmentation, and detection, particularly in contexts such as human behavior recognition and video classification (Simonyan & Zisserman, 2014). The I3D algorithm (Carreira & Zisserman, 2017), grounded in the framework of a three-dimensional convolutional neural network, further refines classification efficiency and network performance by expanding its width. This augmentation enhances the algorithm’s proficiency in comprehending and classifying video data, showcasing notable advancements in the field.

The preceding investigation underscores the limitations inherent in data analysis reliant solely on single modal information. Optimal results are achieved through the fusion of multimodal information and the profound integration of network layers, thus elevating the efficacy of data analysis. Leveraging the capabilities of a 3D convolutional neural network for video features, particularly those with timestamped features, facilitates adept data analysis and feature extraction. Moreover, superior accuracy in analysis is attained through the fusion of multimodal features, specifically sound information, employing decision layer fusion. Therefore, a refined approach involves distinct feature extraction processes for image features and audio data. Subsequently, data analysis is elevated to a more sophisticated level through decision layer fusion, thereby achieving an enhanced and nuanced understanding of the underlying information.

Materials and Methods

Upon scrutinizing the existing landscape of multimodal fusion research, the subsequent analysis dissects the structure of each modal data fusion. Building upon this foundation, our approach entails the initial establishment of dedicated networks for image data feature extraction and audio data extraction. Subsequently, a bespoke module is constructed, ensuring the optimal utilization of these diverse modalities for comprehensive data classification. The intricate details of the model’s specific construction process will be expounded upon in this Section.

I3D-based video feature extraction

The pivotal role of the three-dimensional convolutional neural network (3DCNN) cannot be overstated. Evolving from the foundations of the 2DCNN, the 3DCNN excels in capturing temporal information with heightened efficacy, enabling the effective utilization of temporal intricacies within video data (Al-Hammadi et al., 2020). In the context of video classification, the 3DCNN performs sliding window convolutions on inputs in three dimensions, encompassing both spatial and temporal dimensions. Within the structure of the 3DCNN, the 3D convolutional kernel manifests as a three-dimensional cube. The convolutional operation entails the superimposition of consecutive frames into a 3D sequence, followed by the computation of convolution between this sequence of video frames and the convolution kernel. The resultant feature maps derived from the convolutional layer are then associated with the input video frames, adeptly capturing the nuanced motion information embedded within the video. The 3D convolution operation mirrors its 2D counterpart but extends its application into both spatial and temporal dimensions.

For an input tensor X, a convolution kernel tensor W and bias b, the formulation of 3D convolution is articulated as follows:

(1) Yi,j,k=∑l,m,nXi+l,j+m,k+n⋅Wl,m,n+b

where Yi,j,k are the elements of the output tensor, and l,m,n is the index of the convolution kernel in three dimensions. 3D pooling operations are commonly used to reduce the data size. Maximum pooling and average pooling are two common 3D pooling operations. For maximum pooling, the formula is as follows.

(2) Yi,j,k=maxl,m,nXi×s+l,j×s+m,k×s+n

where Yi,j,k are the elements of the output tensor after pooling, and s is the step size of pooling.

The I3D convolutional neural network, a creation of Google DeepMind Labs, represents a distinctive 3D convolutional neural network derived from the “inflation” of the GoogleNet architecture. This augmentation involves the addition of batch normalization layers after each convolutional operation, mitigating the network’s convergence challenges and alleviating inherent issues related to internal input variability in convolutional neural networks.

In the temporal dimension design of the I3D convolutional neural network, careful consideration is given to preventing the merging of edge information from different objects when the sensory field of the temporal dimension exceeds that of the spatial dimension, and vice versa, ensuring the capture of dynamic scenes (Wang, Koniusz & Huynh, 2019). To achieve this, in the downsampling structure of the I3D network, the design of the first two pooling layers involves setting the step size of the temporal dimension to 1 and maintaining the step size of the spatial dimension at 2. Conversely, in the design of the last pooling layer, the step size of the temporal dimension is set to 2, while the step size of the spatial dimension is set to 7. For the remaining downsampling layers, the step size for both the temporal and spatial dimensions is set to 2. The detailed network structure is illustrated in Fig. 1.

Figure 1 The structure of the I3D.

In contrast to the C3D network, the I3D network incorporates a batch normalization layer into its structure. Leveraging the distinctive network design of GoogleNetV1, this inclusion effectively addresses issues of underfitting caused by excessively large parameters and mitigates training dispersion resulting from internal input variations. When compared to the P3D network, the I3D network showcases a uniquely crafted downsampling structure that enhances the efficiency of extracting feature information from videos. This results in a substantial improvement in the algorithm’s recognition accuracy. Furthermore, in comparison to the T3D network, the I3D network exhibits advantages in both recognition accuracy and speed. The asymmetric downsampling strategy in I3D—using a temporal stride of 1 and spatial stride of 2 in early layers—preserves fine-grained temporal information while reducing spatial redundancy. This design enhances the network’s sensitivity to motion dynamics without significantly increasing computational cost. Temporal downsampling is deferred to later stages to capture long-term dependencies, making this structure more effective than uniform stride configurations such as (2, 2, 2). In this work, we follow the inflation strategy proposed by Carreira & Zisserman (2017), where 2D convolutional filters pre-trained on ImageNet are inflated into 3D kernels by expanding their shape along the temporal dimension. Specifically, a 2D kernel of shape N × N is transformed into a 3D kernel of size T × N × N, where the temporal dimension is initialized either by repeating the 2D weights or scaling them to preserve the original filter response. This inflation mechanism allows the model to benefit from rich spatial representations learned on large-scale image datasets while effectively capturing temporal dynamics through end-to-end training on video data. This transfer learning approach accelerates convergence and improves generalization, particularly under limited video supervision.

MFCC-based audio feature extraction

Mel filtering emulates the nonlinear perception and filtering characteristics of the human auditory system on audio signals. The human auditory system exhibits selective attention to frequencies within an audio signal, with varying sensitivity across different frequency ranges. Notably, the human ear is more attuned to low-frequency signals, displaying enhanced receptivity to these frequencies compared to higher frequencies. The cochlea processes audio signals below 1,000 Hz in a linear fashion, while signals above 1,000 Hz are received in a logarithmic manner (Deng et al., 2020).

To replicate this characteristic, a method involves overlaying a triangular filter bank on the power spectrum. In addition to the cochlear filtering effect, the human ear subjectively and nonlinearly perceives sound. To mirror this subjective perception more closely to that of the human auditory system, the signal is measured using a subjective nonlinear scale known as the Mel scale. This entails converting the frequency domain signal to the Mel scale. The conversion relationship between the Mel scale and the frequency spectrum is illustrated below:

(3) fmel=2595∗log10(1+f700)

(4) f=700(10fmel2595−1).

Equation (3) is the frequency to Mel scale conversion equation, and Eq. (4) is the Mel scale to frequency conversion equation. In these equations, f is the linear frequency in Hertz, fmel is the frequency in the Mel scale, which approximates human auditory perception of pitch.

The function of the Mel filter bank is to mimic the properties of the human ear in the perception of sound frequencies by converting the frequency axes in the spectrogram to the Mel scale.

(5) Hm(k)={0,k<f(m−1)k−f(m−1)f(m)−f(m−1),f(m−1)≤k≤f(m)f(m+1)−kf(m+1)−f(m),f(m)≤k≤f(m+1)0,k>f(m+1)

where f(m) denotes the center frequency of the first m center frequency of the Meier filter. Hm(k) is the weighting coefficient of the m-th Mel filter at the k-th FFT. It defines the triangular filter shape applied to the spectrum.

The Meier filter bank accomplishes the work of smoothing the spectrum, highlighting the resonance peaks of the signal, and eliminating harmonics. At the same time, it reduces the amount of calculation. Because the human ear presents a logarithmic response to sound, the logarithmic energy value is calculated for the output after the power spectrum passes through the Meier filter bank. The logarithmic energy calculation formula is shown in Eq. (6).

(6) s(m)=ln(∑k=0N−1|Xa(k)|2Hm(k))0≤m≤M,

where Xa(k) is the frequency k corresponding to the spectral energy value, obtained directly from the Fourier transform of the audio frame, and Hm(k) is the frequency k is the corresponding Meier energy value, obtained by filtering with a triangular filter. To balance the signal-to-noise ratio (SNR) of the sound signal spectrum, the features extracted by FBank can be normalized using mean normalization, where the average value of each frame is subtracted from all frames. Mean normalization after Meier filtering reduces the computational difficulty while maintaining feature invariance.

Cepstrum is a way of converting a frequency domain signal to the time domain. The cepstrum is obtained by first taking the logarithm of the amplitude value of the standard amplitude spectrum and then visualizing it. The process of obtaining the cepstrum using the discrete cosine transform is shown in Eq. (7):

(7) C(n)=∑m=0N−1s(m)∗cos(πn(m−0.5)M),n=1,2,…,L

where C(n) is the n-th MFCC coefficient, L denotes order of coefficients, and M is the number of diagonal filters, and s(m) denotes the logarithmic energy value of the output of the first m is the number of diagonal filters, and denotes the logarithmic energy value of the output of the first diagonal filter, which is obtained from Eq. (7). This is the standard type-II DCT used in MFCC extraction, which provides energy compaction and decorrelation, and ensures compatibility with prior works in speech and audio processing. The waveform of the cepstrum is similar to the waveform of the spectrum, so when the low frequency in the spectrum has a higher energy value, the low-frequency cepstrum coefficients will also obtain a higher value, and when there are peaks in the spectrum of the high-frequency peaks, the high-frequency cepstrum will also appear peaks. In the spectrum of an audio signal, the frequency corresponding to the higher peak is the main frequency of the audio signal, and the peak is the resonance peak. The main peaks of the audio signal are its characteristics, and the similarity between the spectrum and the cepstrum waveform makes this feature retained. The above process of obtaining the MFCC feature can be represented in Fig. 2 (Gao, 2023).

Figure 2 The MFCC feature extraction process.

Bimodal time series classification model for the audio and image feature fusion

Following the completion of feature extraction utilizing the I3D network for image features and employing MFCC for audio feature extraction, a convergence at the decision level is executed. Subsequently, we introduce the BTSCM network model, designed for the integrated classification of multimedia data. The schematic representation of the BTSCM network model and the procedural framework for feature fusion is illustrated in Fig. 3.

Figure 3 The framework for the BTSCM and proposed bimodal fusion method.

The BTSCM network, as proposed, comprises three primary components: a module for dimensionality reduction of features based on a fully connected network (Lu et al., 2019), a module dedicated to enhancing time series features through LSTM (Shi et al., 2022), and a model augmentation segment based on reinforcement learning using the deep Q network (DQN). The specific amalgamation is depicted in the rightmost section of Fig. 3. Initially, the model integrates with features derived from the extraction network. Subsequently, data undergoes feature extraction and dimensionality reduction via the FCN network. Following this, the enhancement of data is accomplished through the utilization of the LSTM network in conjunction with reinforcement learning, ensuring precise categorization of the data. The subsequent sections will elaborate on each of the three modules.

(1) FCN module: FCN replaces the traditional fully-connected layer with a fully-convolutional layer, enabling the network to accept input images of any size and output predictions of the corresponding size. This enables FCN to process each pixel in an image rather than just classifying the entire image. FCN captures semantic information of an image at multiple scales by integrating different levels of feature maps. This makes the model better adaptable for targets of different sizes and shapes. For the 1DFCN used in this article can be computed by Eq. (8)

(8) E^i,t(l)=f(bi(l)+∑t′=1d⁡⟨Wi,t′,.(l),E.,t+d−t′(l−1)⟩)

where W(l) and b(l) are the weight and biases Etl is a function of the incoming activation matrix f(⋅) is a rectified linear unit. In this article we have chosen FCN32s to accomplish the dimensionality reduction of the data. K is the kernel half-width, r is the dilation rate. This convolution is applied along the temporal dimension, while treating the feature dimension as the input channels. This design enables the 1D-FCN to model both local and long-range temporal patterns depending on the dilation rate. In our implementation, the 1D-FCN is applied along the temporal dimension of the concatenated feature sequence. Specifically, for each video, we first concatenate multi-modal features at the frame level, resulting in a sequence of fixed-length feature vectors over time. The 1D convolutions then operate with kernels that slide along the time axis, while treating the feature dimension as the input channel dimension. This design allows the network to learn temporal dependencies and local patterns across consecutive frames, without mixing information across feature channels.

In this study, we selected FCN32s primarily due to its efficiency in reducing the temporal resolution while retaining high-level semantic representations. FCN32s applies a larger stride in its decoding path, which allows for faster computation and simpler integration with the sequential modeling components (e.g., LSTM). Given the goal of this work is to analyze multi-modal feature interactions rather than fine-grained temporal localization, FCN32s offers a practical balance between performance and complexity. Although variants like FCN8s or FCN16s could provide finer temporal resolution, they significantly increase computational overhead and are more suitable for tasks requiring dense, frame-level predictions. We leave the comparative study of different FCN depths as part of future work focused on architectural optimization.

(2) DQN module: Before the spatio-temporal feature extraction and analysis of LSTM, we add a reinforcement learning module, and in this article, we realize the reinforcement of the model through the DQN network. DQN is a kind of deep reinforcement learning model, which is used for solving the reinforcement learning problem in the discrete action space. It combines the ideas of deep learning and Q-learning, and aims to learn to predict optimal actions in complex environments through neural networks. Reinforcement learning mainly searches for the optimal Q-value through the following Formula (9) to optimize the model.

(9) Q(st,at)=(1−α)⋅Q(st,at)+α⋅(rt+γ⋅maxaQ(st+1,a))

where Q(st,at) denotes the state in which st action is taken in the state at of the Q value, and rt is the value of the action taken in the state st is the value of the immediate reward obtained after taking an action in the state at the immediate reward obtained after the action is taken, the st+1 is the next state transferred to, the α is the learning rate, the γ is the discount factor, and maxaQ(st+1,a) denotes the next state in which the st+1 in which the action is selected a of the maximum Q value.

(3) LSTM module: LSTM is specially designed to solve the long sequence dependence problem, which contains an input gate, a forgetting gate, and an output gate, and through the control of the three gates, it can be a good solution to the gradient explosion problem, and significantly improve the performance of the model. The computational process of its main three gates is shown in Eqs. (10)–(12):

(10) it=σ(Wiixt+bii+Whiht−1+bhi)

(11) ft=σ(Wifxt+bif+Whfht−1+bhf)

(12) ot=σ(Wioxt+bio+Whoht−1+bho)

where xt is the input, and ht−1 is the previous hidden state, and σ denotes the activation function, and W and b are the weights and biases, respectively. We further define the cell state update and the hidden state computation, which are essential components of the LSTM unit. The equations are as follows:

(13) ct=ft⊙ct−1+it⊙c~t

(14) ct=ft⊙ct−1+it⊙c~t

(15) ht=ot⊙tanh(ct)

where c~t is the candidate cell state, ct is the updated cell state, ht is the hidden state (also the LSTM output at time step t), ⊙ denotes element-wise multiplication, tanh is the hyperbolic tangent activation function. These equations define how information is selectively retained, updated, and exposed at each time step, ensuring long-range temporal dependencies are effectively modeled.

In this article we choose a two-layer LSTM network to help complete the data processing and add a fully connected layer to its subsequent to complete the classification. In this article we choose the cross entropy function as the loss function for model tuning.

After further processing of the model features, we add a classification layer after the BTSCM module, which mainly consists of dropout and softmax functions to complete the final classification. In our framework, the DQN serves as a high-level decision-making module that dynamically interacts with the sequential learning process. Specifically, the “action” in the DQN refers to the selection or modulation of attention over feature representations at each time step, effectively guiding the model on which temporal segments or modality features to prioritize. This reinforcement-based mechanism operates in parallel with the LSTM and influences its memory updates by providing external signals that emphasize salient information and suppress redundant or noisy inputs. As a result, the LSTM benefits from a more focused and adaptive encoding of long-term dependencies, thereby improving overall classification accuracy. This integration enables a closed-loop structure where DQN continuously refines the temporal focus based on feedback from the classification performance.

Computing infrastructure

In this study, all experiments were conducted on a high-performance computing workstation running Ubuntu 20.04 LTS. The hardware setup included an Intel Xeon Silver 4214 CPU, an NVIDIA RTX 3090 GPU with 24 GB VRAM, 128 GB DDR4 RAM, and a 2 TB NVMe SSD for fast data storage and access.

The deep learning models were implemented using Python 3.8, with primary support from TensorFlow 2.x and PyTorch 1.11 frameworks. GPU acceleration was enabled using CUDA 11.1 and cuDNN 8.0. Additional libraries such as OpenCV were utilized for image processing, Librosa for audio feature extraction, and Scikit-learn for data manipulation and evaluation metrics. FFmpeg was employed for video and audio decomposition tasks.

Data preprocessing steps

To prepare the video data for classification, we first decomposed each video into a series of frames and corresponding audio streams using FFmpeg tools.

For the image processing pipeline, the extracted frames were resized to a uniform resolution of 224 × 224 pixels to match the input requirements of the I3D (Inflated 3D ConvNet) model. All images were normalized by scaling pixel values to the [0, 1] range to improve training stability and convergence speed.

For the audio processing pipeline, we extracted MFCCs from the audio tracks using Librosa. We computed 13–40 MFCCs per frame with a window size of 25 milliseconds and a stride of 10 milliseconds. These audio features were subsequently standardized by applying mean-zero and unit-variance normalization.

Temporal alignment was carefully performed to ensure that the visual frames and audio features were synchronized based on their timestamps, allowing for consistent multimodal input sequences to the model. To ensure synchronization between audio and video modalities, MFCC features are extracted using a 25 ms Hamming window with a 10 ms hop size, resulting in 100 audio feature frames per second. The video stream, recorded at 25 frames per second (fps), corresponds to a temporal resolution of 40 ms per frame. To align the modalities, we aggregate every four consecutive MFCC frames (covering 40 ms) using average pooling to obtain a single audio feature vector that corresponds to one video frame. This protocol effectively resolves the sampling rate mismatch and enables frame-level audio-visual fusion.

Following feature extraction, the processed image and audio data were formatted as sequential inputs suitable for processing by the BTSCM network, which integrates fully convolutional networks (FCN), DQN, and long short-term memory (LSTM) modules.

Finally, the dataset was divided into training (70%), validation (15%), and test (15%) sets, ensuring that samples from different video categories were proportionally represented in each split.

Experiment

Dataset

Considering the paramount objective of discerning diverse human movement conditions within multimedia data pertaining to art and design, this article conducts its analysis utilizing four databases encompassing human movement video data along with associated audio information: UCF101, Kinetics, Sports-1M, and ActivityNet. UCF101, a widely acknowledged video classification dataset, features 101 distinct action categories, amounting to 13,320 video clips. Kinetics-400, a comprehensive video classification dataset, comprises 400 diverse action categories, totaling nearly 300,000 video clips, rendering it one of the most expansive publicly accessible repositories for video categorization. Sports-1M, an extensive dataset, encompasses 1,133,514 YouTube videos spanning 487 distinct movement categories. ActivityNet, a large-scale video comprehension dataset, encapsulates 203 varied activity categories sourced from YouTube, resulting in a compilation of 10,024 video clips. Despite the expansive nature of these datasets, this article selectively employs a subset for experimentation, owing to constraints imposed by experimental equipment. Within these datasets, video clips commonly consist of sequences of video frames accompanied by corresponding audio clips, aligning seamlessly with the model employed in this study. Throughout the model application process, we initialize the video and undertake a comprehensive analysis of the pertinent data. For all datasets, only video clips containing synchronized audio and clean action labels were retained. To ensure class balance and reduce bias, we manually selected equal numbers of samples per class where feasible. Annotations were taken from the official dataset labels, and no additional manual labeling was performed. This curated subset ensures a manageable yet representative sample for evaluating the system’s performance in multimedia art contexts.

Experiment setup and details

After determining the dataset, we determined the evaluation method of the model, and considering to carry out the classification problem research, we used the classical precision, recall and F1-score metrics for model evaluation, which are defined as shown in Eqs. (16)–(18):

(16) P=TPTP+FP

(17) R=TPTP+FN

(18) F1=2×P×RP+R

where P denotes precision; R denotes recall; F1 denotes F1 coefficient; TP denotes true case; FP denotes false positive case; FN denotes false negative case. Based on the above data and evaluation indexes we carried out the training of the model, and its specific process is shown in Algorithm 1.

Algorithm 1 Training process of BTSCM.

Input: UCF101, Kinetic, Sports-1M, ActivityNet dataset	
Output: Trained BTSCM model	
Initialization: batch size, leaning rate, epochs, weights, bias.	
Define Extracted feature, LSTM, FCN and DQN	
Combine FCN+DQN+LSTM	
while epoch < preset-epoch do	
    Sample a batch of feature.	
    Feed the data to the BTSCM.	
    Update model.	
End	
Parameters tuning	
while epoch < preset-epoch do	
    Feed validation data to BTSCM.	
    Loss and gradients calculation.	
    Model updated.	
    Compute loss	
    Save BTSCM	
end	

To enhance the evaluative process of the models, we have curated a selection of network methodologies capable of integrating video image features and audio features for comparative analysis. The methodologies under consideration include: tensor fusion network (TFN): TFN adeptly models interactions across unimodal, bimodal, and trimodal domains through a three-dimensional Cartesian product, facilitating a sophisticated fusion of feature spaces (Zadeh et al., 2017). Knowledge enriched transformer (KET): The KET model intricately models contextual information employing a hierarchical self-attention module and dynamically introduces external general knowledge into the information flow via a context-aware affective-attention mechanism (Zhong, Wang & Miao, 2019). COSMIC: The COSMIC framework, proposed in this work, is grounded in a general knowledge-oriented approach and relies on an extensive knowledge base, effectively capturing audio features (Ghosal et al., 2020). Bidirectional emotional recurrent unit (BiERU): The BiERU model represents an exceedingly parameter-efficient framework that disregards participant influence. Leveraging a generalized neural tensor block and a two-channel feature extractor, it adeptly captures contextual information (Li et al., 2022).

Experiment results and analysis

Following the culmination of model training, a comprehensive comparative analysis of results across diverse datasets transpired. Deliberating on practical application scenarios and data accessibility, this article meticulously curated a subset of 50 actions from each database for in-depth analysis, culminating in a final classification task. To ensure fair evaluation and avoid class imbalance, the 50 selected actions were chosen to represent a diverse range of motion types (e.g., dynamic, subtle, upper/lower body), while maintaining an approximately equal number of samples per class within each dataset. Selection was guided by the availability of synchronized audio-visual data and the clarity of action labels. This strategy minimizes potential bias in evaluation metrics and ensures comparability across datasets. The outcomes pertaining to the UCF101 dataset are meticulously delineated in Table 1 and visually depicted in Fig. 4.

Table 1 The comparison result on the UCF101 dataset.

Method	Precision	Recall	F1-score	
TFN	0.889	0.849	0.868	
KET	0.876	0.827	0.851	
COSMIC	0.892	0.801	0.844	
BiERU	0.853	0.822	0.837	
Ours	0.901	0.893	0.897	

Figure 4 The comparison result on the UCF101 dataset.

The discernment across the three indicators readily reveals the commendable recognition efficacy of the methods introduced in this article. The model exhibits a nuanced equilibrium in performance. It is noteworthy, however, that the superior quality and reduced noise within the dataset contribute to the heightened results observed in multi-classification tasks. For a more expansive evaluation, the outcomes pertaining to the other two datasets, Kinetics and Sports-1M—widely employed in the multi-classification study of human motor behavior—are meticulously presented in Tables 2 and 3, complemented by the graphical representation in Figs. 5 and 6.

Table 2 The comparison result on the kinetics dataset.

Method	Precision	Recall	F1-score	
TFN	0.598	0.601	0.600	
KET	0.602	0.615	0.608	
COSMIC	0.625	0.639	0.632	
BiERU	0.612	0.592	0.602	
Ours	0.649	0.637	0.643	

Table 3 The comparison result on the Sports-1M dataset.

Method	Precision	Recall	F1-score	
TFN	0.588	0.539	0.562	
KET	0.601	0.587	0.594	
COSMIC	0.587	0.561	0.574	
BiERU	0.592	0.602	0.597	
Ours	0.615	0.589	0.602	

Figure 5 The comparison result on the Kinetics dataset.

Figure 6 The comparison result on the Sports-1M dataset.

Given the extensive diversity within the dataset and the current state-of-the-art (SOTA) algorithms struggling to surpass a classification performance threshold of 70%, it is noteworthy that the recognition rate achieved by the method proposed in this article, while lower, still surpasses that of existing multimodal fusion methods. Figure 6 illustrates that the Recall of the Bi-ERU method marginally surpasses that of the proposed method, yet the overall performance disparity is minimal. This observation substantiates the assertion that the proposed method exhibits a more balanced performance. The comparison result on the ActivityNet dataset is shown in Table 4 and Fig. 7.

Table 4 The comparison result on the ActivityNet dataset.

Method	Precision	Recall	F1-score	
TFN	0.764	0.701	0.731145	
KET	0.759	0.723	0.740563	
COSMIC	0.801	0.786	0.793429	
BiERU	0.821	0.801	0.810877	
Ours	0.847	0.829	0.837903	

Figure 7 The comparison result on the ActivityNet dataset.

Building upon this foundation, an analysis was conducted on the ActivityNet dataset. Given the dataset’s clarity, the overall performance surpasses that of the aforementioned two datasets, with precision reaching 84.7%, notably outperforming other methods. Building upon this success, an analysis of data from the forthcoming art platform was undertaken. The platform’s data stream, characterized by a lower volume, predominantly comprises uploaded short videos and related audio. Selecting pertinent clips for testing, the results for various multimodal data analysis methods are vividly illustrated in Fig. 8.

Figure 8 The result for the method comparison on the self-established dataset.

As shown in Fig. 8, we still compared four types of networks fusing audio and image information such as TFN, and the results showed that all types of methods achieved more than 85% precision, while the network proposed in this article achieved better results under all three types of indexes, and its results showed that the method proposed in this article has a better equilibrium performance, and improves the recognition accuracy to a certain extent. After completing the model analysis, we carried out the ablation experiment of the reinforcement learning module, and its results are shown in Table 5 and Fig. 9.

Table 5 The statistical result in Fig. 9.

Index	RL	No RL	
Precision	0.895 ± 0.021	0.744 ± 0.013	
Recall	0.900 ± 0.018	0.719 ± 0.014	
F1-score	0.897 ± 0.006	0.731 ± 0.007	

Figure 9 Comparison of ablation experiments of reinforcement learning.

The mean precision, recall and F1-score and the related standard deviation for reinforcement learning influence is also represented as follows:

In the conducted ablation experiments, the model underwent testing both with and without the incorporation of reinforcement learning on the self-established dataset. The experimental outcomes reveal that the inclusion of the reinforcement learning module enhances the recognition efficacy of the LSTM feature extraction layer employed in this study. Across various batch sizes, the effect observed in the model with the reinforcement learning module surpasses that of the model without, affirming the robustness of the BTSCM method and its ability to achieve superior recognition outcomes.

Discussion

In the context of intelligent auditing and video classification for art creation platforms, this article introduces a bimodal video recognition network, the BTSCM framework, which adeptly integrates image and sound information. The robustness of this framework’s recognition effect surpasses that of established methods such as TFN and KET. Notably, the BTSCM method employed in this study demonstrates advantages over TFN, which utilizes the Cartesian product to model interactions among unimodal, bimodal, and trimodal elements. Furthermore, the network complexity in our approach is comparatively lower than that of the KET-based transformer method. At this juncture, enhancing the robustness of modal fusion algorithms remains a pivotal developmental avenue in this research domain. The BTSCM model, as delineated in this article, leverages reinforcement learning to focus on analyzing specific preference classes of data within the framework. This nuanced approach contributes to better balancing the classification performance of the model across diverse classes. The method consistently outperforms other frameworks in practical dataset tests, attesting to the rationality of the proposed framework.

In the contemporary landscape of visual communication, delving into the humanistic environment and background of current design status quo is imperative. Analyzing the evolution and inheritance of design forms from the perspective of cultural ontology in visual communication design is essential for clarifying the influence of culture on design activities. This exploration lays the groundwork for constructing a value system for new-age design culture that aligns with societal development and embraces advanced cultural concepts. The globalization of design and production activities has streamlined information transmission and elevated design culture standards. However, despite increased efficiency in material production, the homogenization of design forms fails to meet the diverse cultural needs at the spiritual level. The innovation in visual communication design emanates from innovative design thinking guided by cultural influence. The burgeoning array of visual expression forms facilitated by technological advancements necessitates a reconfiguration and integration of visual communication design elements with post-modern design culture through innovative design thinking. Therefore, a comprehensive analysis of artworks on art platforms from the perspective of art communication, coupled with intelligent auditing and evaluation, is indispensable for the future healthy development of art platforms.

Conclusion

This article introduces the BTSCM video content classification network, designed to process bimodal information from video images and sound. The network employs the I3D network for video image feature extraction and MFCC for audio feature extraction. The self-constructed BTSCM module is then utilized for video content classification, integrating different levels of feature maps through FCN to capture semantic information at various scales. The BTSCM module further employs the LSTM method with a reinforcement learning module for deep feature fusion, enhancing classification accuracy. Multiple datasets are employed for testing, and while the model falls short of achieving the state-of-the-art (SOTA), it outperforms traditional multimodal fusion networks across three metrics. Notably, in tests on a self-built dataset, the model achieves a classification precision exceeding 90% for common works, surpassing other traditional methods. These results underscore the framework’s efficacy for future content review and classification methods for art platforms, providing valuable technical support.

Future research endeavors aim to enhance the generalization performance of the current model by broadening its applicability to a wider array of videos and content. Additionally, efforts will be directed towards integrating more multimedia information, including scripts, subtitles, and other relevant data. While the BTSCM model demonstrates commendable performance in human action datasets, further testing is warranted for the study of artworks encompassing animation and hand-drawing.

Limitations

The proposed BTSCM framework demonstrates strong classification performance, several limitations must be acknowledged. The model was trained and tested on a dataset, which may not fully represent the diversity and complexity of real-world multimedia art content, potentially limiting generalizability. Additionally, the current multimodal fusion strategy, though effective, may not fully capture deeper cross-modal interactions that more advanced architectures like attention-based models could exploit. The framework’s computational complexity also poses challenges for real-time deployment on resource-constrained devices.

Supplemental Information

Supplemental Information 1 Data.

Supplemental Information 2 README.

Supplemental Information 3 Code.

Additional Information and Declarations

Competing Interests

The author declares that she has no competing interests.

Author Contributions

Qing Yun conceived and designed the experiments, performed the experiments, analyzed the data, performed the computation work, prepared figures and/or tables, authored or reviewed drafts of the article, and approved the final draft.

Data Availability

The following information was supplied regarding data availability:

The UCF101 dataset is available at Zenodo: Mikolaj Buchwald. (2023). Keras video classification example with a subset of UCF101—Action Recognition Data Set (top 5 videos) (1.2) [Data set]. Zenodo. https://doi.org/10.5281/zenodo.7924745.

The Kinetics dataset is available at: https://paperswithcode.com/dataset/kinetics, doi: 10.1109/TPAMI.2020.2992393.

The Sports-1M dataset is available at Kaggle: https://www.kaggle.com/datasets/sabahesaraki/sports-1m-dataset, doi: 10.1109/CVPR.2014.223.

The ActivityNet is available at Zenodo: Guo Chen. (2022). TSN feature on ActivityNet v1.3 [Data set]. Zenodo. https://doi.org/10.5281/zenodo.6650813.

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
