# Peer review of "Parametric art creation platform design based on visual delivery and multimedia data fusion"

_PeerJ Computer Science, doi:10.7717/peerj-cs.3175_

## Round 0.1 · original submission · Major Revisions

Please see both detailed reviews. The reviews highlight that while the manuscript presents a novel bimodal classification model (BTSCM) with a well-structured approach, it suffers from several technical weaknesses. Key issues include incomplete or ambiguous mathematical formulations (e.g., 3D convolution, LSTM gates, MFCC DCT), insufficient experimental details (e.g., dataset balancing, alignment protocols, ablation studies), and missing validation metrics (e.g., statistical significance, confusion matrices). The reviewers also note a lack of clarity in architectural choices (e.g., asymmetric pooling strides, DQN's role) and call for comparative analyses to justify design decisions. Addressing these gaps would strengthen the paper's reproducibility, validity, and impact.

**Language Note:** The review process has identified that the English language must be improved. PeerJ can provide language editing services - please contact us at [email protected] for pricing (be sure to provide your manuscript number and title). Alternatively, you should make your own arrangements to improve the language quality and provide details in your response letter. – PeerJ Staff

Reviewer 1 ·

Basic reporting

The proposed method has merit, but a significant revision is necessary to address technical ambiguities.
 The manuscript mentions various pooling strides across layers (e.g., temporal stride = 1, spatial = 2 for early layers). However, no ablation or analysis is provided to validate why this asymmetric sampling improves temporal sensitivity. A comparative test against uniform stride configurations (e.g., (2,2,2)) would be illuminating.

Experimental design

The equation includes undefined variables such as “the center frequency of the first Meier filter” — possibly a typographical error for “Mel filter.” Additionally, the precise construction of the triangular filter bank (e.g., number of bands, bandwidth overlap) is missing.
 The equation does not define variables like CnC_nCn, NNN, or the exact form of the DCT. Please provide a full expression of the DCT formula used (type-II or type-III), as this affects MFCC reproducibility and comparability with other work.
 The manuscript briefly states that MFCC features are synchronized with video frames, but omits the alignment protocol. Are MFCCs aggregated across fixed frame windows (e.g., 30 ms per frame)? How are unequal sampling rates between video (e.g., 25 fps) and audio (e.g., 16 kHz) resolved?

Validity of the findings

Equation (8) for 1D FCN is vague. How is this applied to temporal sequences? Are the convolutions over time, feature dimensions, or both?
 The notation does not match standard FCN terminology (e.g., use of kernel width, dilation). Please re-specify with complete dimensions.

Additional comments

No details are provided about the number of videos, duration per clip, annotation process, or class distribution. This makes it difficult to assess the generalizability or real-world applicability of the system to art platforms.
 The paper uses precision, recall, and F1-score but omits accuracy and class-wise confusion matrices, which are especially important for unbalanced or multi-class tasks.
 The model uses a 2-layer LSTM and FCN32s, but no explanation or alternatives are tested. Varying these architectures (e.g., FCN8s vs FCN32s) may yield insights into optimal depth/width trade-offs.

Reviewer 2 ·

Basic reporting

The manuscript presents a novel Bimodal Time Series Classification Model (BTSCM) that integrates I3D, MFCC, FCN, LSTM, and DQN to perform video-based multimodal classification, particularly for art platform applications. The work is well-structured and combines state-of-the-art techniques in feature extraction and fusion. However, several technical weaknesses, ambiguities, and missing justifications reduce the strength of the contribution.
The 3D convolution formula is referenced but not fully expressed. Please explicitly define the indexing and tensor dimensions of inputs, weights, and outputs. Without a complete form, readers cannot reconstruct the implementation or understand the kernel’s temporal behavior.
Although Figure 1 is referenced, the exact inflation mechanics from 2D to 3D (i.e., filter reuse across time) are not specified. It would be valuable to clarify whether pre-trained 2D kernels are used and then inflated (as in Carreira & Zisserman, 2017), or if the model is trained from scratch.

Experimental design

Equations (10)–(12) omit several key terms. The forget gate and cell state equations, which are core to LSTM, are not provided. This reduces clarity regarding how the temporal states evolve.
It's unclear whether the LSTM operates on concatenated audio+visual embeddings or processes modalities sequentially.
The paper selects 50 actions from each dataset but does not explain the criteria or whether these subsets are balanced across classes. This could lead to bias in reported performance metrics.

Validity of the findings

The paper should also report statistical significance (e.g., standard deviation across runs or confidence intervals) to validate performance improvements.
Figure 9 suggests performance gains from DQN, but the mechanism remains unclear. What does the DQN “action” represent in this classification context? How does it enhance the LSTM’s memory updates?
Please provide numerical values or significance testing for the ablation experiments. Bar plots alone are insufficient.

---

## Round 0.2 · accepted · Accept

Both reviewers have confirmed that the authors have addressed their comments.

Reviewer 1 ·

Basic reporting

no comment

Experimental design

no comment'

Validity of the findings

no comment'

Additional comments

no comment'

Reviewer 2 ·

Basic reporting

Overall look after revision is better.

Experimental design

The revised version looks well and to be accepted

Validity of the findings

Paper looks more compact now.